# Novel Multitarget Therapies for Lung Cancer and Respiratory Disease

**DOI:** 10.3390/molecules25173987

**Published:** 2020-09-01

**Authors:** Masako Yumura, Tatsuya Nagano, Yoshihiro Nishimura

**Affiliations:** Division of Respiratory Medicine, Department of Internal Medicine, Kobe University Graduate School of Medicine, Kobe 650-0017, Japan; y_ym_mapie82@yahoo.co.jp (M.Y.); nishiy@med.kobe-u.ac.jp (Y.N.)

**Keywords:** multitarget, pemetrexed, ALK inhibitors, angiogenesis inhibitors

## Abstract

In recent years, multitarget drugs for neurological diseases such as Alzheimer’s disease have been developed and well researched. Many studies have revealed that multitarget drugs are also useful for lung cancer and respiratory diseases. Pemetrexed is a multitargeted antifolate with strong antitumor activity against mesothelioma and lung adenocarcinoma. Crizotinib is an ATP-competitive tyrosine kinase inhibitor that targets *c-MET*, *ROS1*, and *ALK*. Alectinib is known as an *ALK* inhibitor but also targets *LTK*, *CHEK2*, *FLT3*, *PHKG2*, and *RET*. Sorafenib is a tyrosine kinase inhibitor that targets RAF kinase, *KIT*, *VEGFR*, *PDGFR1β*, *FLT3*, and *RET*. Nintedanib is a multiple tyrosine kinase inhibitor that targets *FGFR*, *PDGFR*, and *VEGFR*. In this review, we summarize the mechanisms of action of multitarget therapies and report the results of the latest clinical trials.

## 1. Introduction

Important discoveries of new drugs have been made based on the strategy of targeting one gene with one drug in one disease [1]. This strategy is considered important to prevent the disadvantages of accidental targeting of other substances. Accordingly, drugs that interact with multiple targets have long been considered undesirable, partly because they have been associated with adverse side effects. However, owing to recent discoveries indicating the complexity of intractable diseases such as cancer and neurological diseases, single-target drugs are thought to not be sufficiently effective, and since early 2000, multitarget drugs have been rapidly developed [2]. Multitarget drugs have synergistic effects as they exhibit different modes of action, which lead to improved adherence, because the number of drugs administered to patients can be reduced. For example, because the combination of venlafaxine and fluoxetine to treat depression increases the side effects of anticholinergic activity, a multitarget drug may lead to reduced side effects [3,4,5].

Recently, multitargeted ligands have been studied in various diseases such as Alzheimer’s disease, depression, poisoning, glaucoma, and nonalcoholic steatohepatitis (NASH) [6]. In this review, we summarize the mechanisms of action of multitarget therapies and the results of the latest clinical trials and introduce novel compounds and discuss the limitations of multitarget drugs.

## 2. Pemetrexed

Pemetrexed is a folate antimetabolite that exhibits strong and broad antitumor activity by inhibiting multiple folate-metabolizing enzyme pathways. Pemetrexed is mainly taken up into cells by the reduced folate carrier (RFC) and undergoes polyglutamine oxidation by folyl polyglutamate synthase (FPGS). When pemetrexed is subjected to polyglutamine oxidation, its intracellular retention is increased, and its affinity for certain folate-metabolizing enzymes is also increased. Pemetrexed and its polyglutamates inhibit multiple folate-metabolizing enzymes involved in thymine and purine nucleotide biosynthetic pathways, such as thymidylate synthase (TS), dihydrofolate reductase (DHFR), and glycinamide ribonucleotide formyltransferase (GARFT), and thus cause imbalance in the cellular nucleotide pool, inhibit DNA and RNA synthesis, and induce growth inhibition and cell death [7] (Figure 1 [8,9]).

Pemetrexed is currently used for malignant pleural mesothelioma and unresectable advanced/recurrent non-small cell lung cancer (NSCLC). Recently, low TS expression was reported to improve the therapeutic effect of chemotherapies including pemetrexed in NSCLC patients [10,11], and it is thought that further research will allow pemetrexed to be used for tailored treatment.

Malignant mesothelioma, which arises from the mesothelial cells that line the inner surface of the chest cavity, is associated with asbestos inhalation. A phase II study of pemetrexed alone showed a response rate (RR) of 14.1% and a median survival time (MST) of 10.7 months [12]. This result is better than that obtained with cisplatin [13] or gemcitabine [14]. A subsequent phase III trial comparing cisplatin alone and combination therapy with cisplatin and pemetrexed (pemetrexed/cisplatin) was conducted in 20 countries, including the United States and countries in Europe, and revealed that survival with pemetrexed/cisplatin treatment was superior to that with cisplatin alone (MST of 12.1 months vs. 9.3 months in the pemetrexed/cisplatin arm and cisplatin alone arm, respectively; *p* = 0.020). The median time to progression was significantly longer in the pemetrexed/cisplatin arm than in the cisplatin alone arm (5.7 months vs. 3.9 months, *p* = 0.001). The RR in the pemetrexed/cisplatin arm was higher than that in the cisplatin alone arm (41.3% vs. 16.7%, *p* < 0.0001) [15]. Additionally, in this trial, a relationship between folic acid and vitamin B12 deficiencies and the occurrence of severe toxicity graded by the Common Terminology Criteria for Adverse Events (CTCAE, grade 4 myelosuppression, grade 3/4 diarrhea, mucositis, infection, etc.) was found, and supplementation with folic acid and vitamin B12 reduced toxicity. Therefore, beginning in the middle of the test, investigators used supplemental vitamin B12 and folic acid. Pemetrexed/cisplatin was subsequently approved by the US Food and Drug Administration (FDA) in February 2004 and is now the standard treatment for malignant mesothelioma. In addition, the RR was 32%, and the MST was promising at 451 days in a phase I study of pemetrexed/carboplatin [16]. In a phase II trial of pemetrexed/carboplatin, the RR ranged from 18.6% to 25%, and the MST was 12.7–14.1 months, showing favorable results. Thus, pemetrexed/carboplatin is a treatment option against malignant mesothelioma [17,18].

A phase III study comparing pemetrexed and docetaxel in lung cancer conducted in patients who were previously treated with platinum-based chemotherapy for NSCLC revealed that pemetrexed has an efficacy equivalent to that of docetaxel but is less toxic than docetaxel [19]. Pemetrexed was then approved as a second-line treatment for NSCLC in the US in August 2004 and in Europe, in September 2004. Another phase III trial comparing pemetrexed/cisplatin with gemcitabine/cisplatin in chemo-naïve NSCLC patients (JMDB trial) showed a prolonged MST in the pemetrexed/cisplatin group with a significant difference in MST between patients with non-squamous cell carcinoma and squamous cell carcinoma [20]. Based on this result, pemetrexed/cisplatin was approved in Europe in April 2008 and in the US, in September 2008 as a first-line treatment for NSCLC. Furthermore, a double-blind, randomized phase III trial (PARAMOUNT trial) was conducted to evaluate the efficacy of maintenance therapy with pemetrexed after induction therapy with pemetrexed/cisplatin. This trial was conducted in patients who were untreated for stage IIIB/IV NSCLC and responded to four courses of induction therapy with pemetrexed/cisplatin. The results showed a significantly prolonged progression-free survival (PFS) after maintenance therapy compared with placebo therapy (4.4 months vs. 2.8 months, respectively; hazard ratio (HR): 0.62, 95% confidence interval (CI): 0.50–0.73, *p* < 0.0001) and significantly better overall survival (OS) in the maintenance group than the placebo group (13.9 months vs. 11.0 months, respectively; HR: 0.78, 95%CI: 0.64–0.96, *p* = 0.0195). The patient’s quality of life (QOL) was not reduced despite grade 3/4 anemia (pemetrexed group: 6.4% vs. placebo group: 0.6%); neutropenia (5.8% vs. 0%); fatigue (4.7% vs. 1.1%); leukopenia (2.2% vs. 0%); nausea (0.6% vs. 0%); or vomiting (0.3% vs. 0%). Based on this study, after four courses of combination pemetrexed/cisplatin therapy, continuing maintenance therapy with pemetrexed is recommended for patients with no disease progression and acceptable toxicity [21]. Recently, a phase III trial was conducted to evaluate the efficacy of addition of pembrolizumab, an immune checkpoint inhibitor, to pemetrexed/platinum-based drugs in patients with PS 0-1 stage IV NSCLC without *epidermal growth factor receptor* (*EGFR*) mutation or *anaplastic lymphoma kinase* (*ALK*) translocation (KEYNOTE-189) [22]. In the interim analysis, PFS and OS, the primary endpoints, showed an HR of 0.52 (8.8 months vs. 4.9 months, 95%CI: 0.43–0.64, *p* < 0.0001) and an HR of 0.49 (median not reached vs. 11.3 months, 95%CI: 0.38–0.64, *p* < 0.0001), respectively. Table 1 summarizes the results of clinical trials of pemetrexed (Table 1).

## 3. Crizotinib

ALK is a cell membrane protein with a transmembrane domain, and human ALK consists of 1620 amino acids. Since ALK contains a tyrosine kinase domain in its intracellular region, it is thought to belong to the receptor tyrosine kinase family, the members of which are activated in response to extracellular stimulation. ALK is an orphan member of the insulin superfamily of receptor tyrosine kinases (RTKs), which are normally expressed in only the central nervous system, small intestine, and testis [23,24]. In 1994, it was reported that the *nucleophosmin ALK* (*NPM1*-*ALK*) fusion gene was present on the t (2; 5) reciprocal translocation in anaplastic large cell malignant lymphoma, and in 2007, a fusion protein consisting of *echinoderm microtubule-associated protein like-4* (*EML4*) and *ALK* was found in 6.7% of NSCLC patients [25]. *ALK*-positive NSCLC patients were mostly non-smokers or light smokers and relatively young, had adenocarcinoma, and generally did not exhibit gene mutations such as the *EGFR* and *Kirsten rat sarcoma viral oncogene homolog* (*KRAS*) mutations [26].

Crizotinib is an ATP-competitive tyrosine kinase inhibitor (TKI) that selectively inhibits the activities of anaplastic lymphoma kinase (*ALK*); *c-MET*/hepatocyte growth factor receptor (*HGFR*); and c-ROS oncogene 1 (*ROS1*) and their oncogenic variants (*ALK* fusion protein, *c-MET*/*HGFR* variant, and *ROS1* fusion protein), leading to inhibition of phosphoinositide 3 kinase (PI3K)/v-act murine thymoma viral oncogene homolog (AKT)/mammalian target of rapamycin (mTOR) or rat sarcoma protein (RAS)/v-raf murine viral oncogene homolog (RAF)/mitogen-activated protein kinase (MAPK)/extracellular signal regulated kinase (ERK) kinase MAPK (MEK)/MAPK signaling (Figure 2). The *ALK* fusion protein and some *ROS1* fusion proteins are constitutively activated by dimerization and activate many downstream signaling factors to promote the cell cycle, proliferation, and survival [27]. Crizotinib is believed to exhibit antitumor effects by inhibiting the kinases *ALK*, *ROS1*, and *c-MET* and suppressing the activation of these factors, tumor cell proliferation, and tumor angiogenesis.

An overseas phase I study (PROFILE 1001) [28] and an international joint phase II study (PROFILE 1005) were conducted in patients with *ALK*-positive NSCLC and showed favorable results by the fluorescence in situ hybridization (FISH) method. Then, an international phase III study (PROFILE 1007) conducted in previously treated *ALK*-positive NSCLC patients to compare crizotinib with conventional standard treatments using docetaxel or pemetrexed showed a significantly prolonged PFS [29]. Further, even for patients with untreated *ALK*-positive NSCLC, a strong antitumor effect with an RR of 74% and a median PFS of 10.9 months was observed (PROFILE 1014) [30]. Based on these results, crizotinib became a first-line ALK inhibitor for use in *ALK*-positive NSCLC. Although most of the adverse events such as digestive symptoms and visual impairment were grade 1, other serious adverse events such as interstitial pneumonia, liver injury, and QT prolongation were also reported. Notably crizotinib acts as a TKI of ALK as well as ROS1 [31]. In the expanded cohort of the PROFILE 1001 trial, three out of 50 patients with *ROS1*-positive NSCLC had a complete response, as determined by the Response Evaluation Criteria in Solid Tumors, and 33 patients had a partial response with an RR of 72% (95%CI: 58–84%), median response time of 17.6 months (95%CI: 14.5 months to not reached), and a median PFS of 19.2 months (95%CI: 14.4 months to not reached) [32]. Based on these results, the expanded use of crizotinib for *ROS1*-positive NSCLC was approved by the US FDA in March 2016 and the European EMA, in August of the same year. A clinical phase II trial of crizotinib for *ROS1*-positive NSCLC was conducted in four East Asian countries—Japan, China, South Korea, and Taiwan. A total of 127 patients were enrolled, and among these patients, 17 showed complete response and 74 showed partial response, with an RR of 71.7% (95%CI: 63–79.3%). The median duration of response was 19.7 months (95%CI: 14.1 months to not reached), and the median PFS was 15.9 months (95%CI: 12.9–24 months), demonstrating the high efficacy of crizotinib [33]. Based on these results, an application to extend the use of crizotinib to *ROS1* fusion gene-positive unresectable advanced recurrent NSCLC was filed in Japan.

## 4. Alectinib

Alectinib, an ALK inhibitor developed in Japan, is a low-molecular-weight compound that is also effective for cell lines with the L1196M (gatekeeper) and C1156Y mutations, which have been implicated in resistance to crizotinib [34]. From the results of crystal structure analysis, it was confirmed that alectinib binds the AFG-binding site DFG-in of ALK. Alectinib was originally regarded as an ALK-TKI with a high selectivity for *ALK*, but Kodama et al. investigated 451 biochemical kinases and found that in addition to *ALK* and *leukocyte receptor tyrosine kinase (LTK)*, *checkpoint kinase* (*CHEK2*), *Fms-like tyrosine protein kinase* (*FLT3*) (D835Y), *phosphorylase kinase gamma submit 2* (*PHKG2*), *proto-oncogene ret* (*RET*), and *RET* (M918T) were also inhibited. In particular, RET kinase activity was strongly inhibited, leading to blocking PI3K or RAS signaling [35]. Notably, since alectinib does not block vascular endothelial growth factor receptors (VEGFR2), unlike multikinase inhibitors, alectinib has fewer side effects associated with its antiangiogenetic properties [36].

Phase I/II clinical trials of alectinib include the AF-001JP trial, targeting crizotinib in untreated cases, and the AF-002JG trial, targeting crizotinib in treated cases. Both of these were single-group trials, and these trials showed high RRs of 93.5% in untreated cases and 55% in previously treated cases [37,38]. In the J-ALEX trial, a phase III trial conducted in Japan, crizotinib was compared with alectinib as a first-line treatment, which showed an RR of 92% in the alectinib group vs. 79% in the crizotinib group [39]. The final report showed a median PFS of 10.2 months with crizotinib vs. 34.1 months with alectinib (HR: 0.37, 95%CI: 0.26–0.52) in 2019. Furthermore, PFS was significantly improved in the alectinib group compared to the crizotinib group [40]. Subsequently, in an international phase III clinical trial (ALEX trial), crizotinib and alectinib were compared as first-line treatments for *ALK*-positive NSCLC. The median PFS was 11.1 months in the crizotinib group, and the median PFS was not reached in the alectinib group (HR: 0.47, 95%CI: 0.34–0.65) [41]. The results of the J-ALEX study showed that the main adverse events that occurred with alectinib were constipation, nasopharyngitis, and dysgeusia, and fewer adverse events of grade 3 or higher were reported in the alectinib group (36.9%) than in the crizotinib group (60.6%). Table 2 summarizes the results of clinical trials of crizotinib and alectinib (Table 2).

## 5. Sorafenib

Angiogenesis consists of multiple processes including sprouting, invasion, migration, proliferation, lumen formation, and the maturation of endothelial cells from existing vascular endothelial cells and vascular endothelial progenitor cells, each of which is regulated by single or multiple angiogenic factors. In addition, the extracellular matrix, adhesion molecules, and various proteases have important functions in the process of angiogenesis [42]. Tumors measuring up to a few millimeters in size can acquire the oxygen and nutrients they need by spreading from their environment. However, when the size of the tumor becomes large, blood vessels (tumor blood vessels) that supply oxygen and nutrients to the tumor are required [43]. Many tumors secrete angiogenic factors to build tumor blood vessels; among the many angiogenic factors, the VEGF family is the most studied. The VEGF signaling pathway consists of three subtypes of receptor tyrosine kinases (VEGFR-1, VEGFR-2, and VEGFR-3); five subtypes (VEGF-A, VEGF-B, VEGF-C, VEGF-D, and VEGF-E); and two subtypes of placental growth factor (PlGF-1 and P1GF-2) as ligands [44].

The angiogenic effect of VEGF is thought to be exerted mainly through VEGFR2 on the vascular endothelium, and VEGF is a target molecule for many drugs. Specifically, it is thought to promote growth, survival, acquisition of migration ability, and acquisition of invasion ability of vascular endothelial cells, and a new blood vessel for the tumor is created [45]. Furthermore, VEGF enhances the permeability of existing blood vessels to create a microenvironment in which vascular endothelial cells can easily migrate and promote the chemotaxis of vascular endothelial cells and progenitor cells of pericytes to promote tumor angiogenesis. As in other carcinomas like colon cancer and gastric cancers, overexpression of *VEGF* is a poor prognostic factor for lung cancer [46]. Tumor angiogenesis inhibitors can be broadly classified into two groups: drugs that inhibit the binding of *VEGF-A* and *VEGFR-2* and multikinase inhibitors, which are small-molecule compounds that inhibit the kinase activity of VEGFR. The *VEGF-A* or *VEGFR-2* inhibitors that have been approved for the treatment of NSCLC in Japan include bevacizumab, which binds *VEGF-A* and inhibits its activity, and ramucirumab, which binds VEGFR-2 and inhibits its activity. Pazopanib, regorafenib, sorafenib, sunitinib, and nintedanib among others are known to be multikinase inhibitors [47,48,49]. Pazopanib is used in soft tissue tumors and renal cell carcinoma; regorafenib is used in rectal cancer and gastrointestinal stromal tumors (GISTs); and sunitinib is used in GIST, renal cell carcinoma, and pancreatic neuroendocrine tumors.

Sorafenib is a multitargeted TKI that targets v-raf murine viral oncogene homolog (RAF) kinase, c-*KIT*, *VEGFR*, platelet-derived growth factor receptors *(PDGFR)-1β*, *FLT-3*, and *RET* leading to inhibition of PI3K or RAS signaling [50,51]. It is currently approved by the FDA for renal cell cancer (2005), hepatocellular carcinoma (2007), and thyroid cancer (2013). Adverse events of sorafenib include hypertension, skin disorders (particularly hand-foot syndrome), liver disorders, elevated lipase and amylase levels, and interstitial pneumonia. In NSCLC, a single-agent phase II clinical trial of sorafenib was conducted in 54 previously treated patients with a schedule of 400 mg of sorafenib twice daily. Although neither complete nor partial response was observed in 51 evaluable patients, stable disease was observed in 30 patients (58.5%), of which 15 (28.8%) also showed tumor shrinkage. The median PFS was 2.7 months, and the median OS was 6.7 months. The main adverse events of grade 3 or higher were hand-foot syndrome, hypertension, fatigue, and diarrhea. Death from pulmonary hemorrhage was observed in one patient with squamous cell carcinoma [52]. Subsequently, Paz-Ares et al. published the MISSION study confirming the efficacy of sorafenib as a third/fourth-line treatment in patients with advanced and recurrent NSCLC. Although PFS was clearly prolonged with sorafenib compared to placebo, the OS did not change. PFS was prolonged in both patients with wild-type *KRAS* and patients with *KRAS* mutations, but OS was unchanged and has not been clinically used [53]. To examine sorafenib as a combination therapy with an existing standard treatment regimen, a phase III clinical trial (ESCAPE trial) for untreated NSCLC was conducted to evaluate the effect of the addition of sorafenib to paclitaxel/carboplatin therapy. However, because the interim analysis did not show extension of the OS, which was the primary endpoint, and an increase in mortality was observed in squamous cell carcinoma, the trial was terminated early [54].

## 6. Nintedanib

Nintedanib is an antifibrotic drug that inhibits multiple tyrosine kinases, including fibroblast growth factor receptors (*FGFRs*), *PDGFRs*, and *VEGFRs*, including *PDGFRα*, *PDGFRβ*, *FGFR1*, *FGFR12*, *VEGFR1*, *VEGFR2*, and *VEGFR3*, leading to PI3K, RAS, or focal adhesion kinase (FAK)/paxillin signaling (Figure 3) [55,56]. Nintedanib also inhibits nonreceptor kinases such as *FLT-3*, *RET*, lymphocyte-specific tyrosine kinase (*LCK*), tyrosine-protein kinase lyn (*LYN*), and proto-oncogene tyrosine protein kinase src (*SRC*) [57].

Through inhibition of fibrotic growth factor receptor as described above, the progress of fibrosis is expected to be delayed by nintedanib, and nintedanib is used as a therapeutic drug for idiopathic pulmonary fibrosis (IPF). In a phase II trial in IPF patients, a decrease in forced vital capacity (FVC), which was the primary endpoint, was suppressed in the nintedanib group compared with the placebo group (FVC: −0.06 L per year vs. −0.19 L per year, *p* = 0.06). The frequency of acute exacerbation, which was the secondary endpoint, was significantly lower in the nintedanib group than in the placebo group (2.4% per year vs. 15.7% per year) [58]. Following this trial, two phase III trials (INPULSIS-1 and INPULSIS-2 trial) were conducted [59,60]. The primary endpoint was the annual FVC decline rate (mL/year), and important secondary endpoints were time taken to the first acute exacerbation of IPF at 52 weeks (reported by the investigator) and changes in the total St. George’s Respiratory Questionnaire (SGRQ) score at 52 weeks from baseline. As a result, the adjusted annual change in FVC was significantly lower in the nintedanib group than in the placebo group (*p* < 0.001) for both trials. In the INPULSIS-1 trial, no significant difference in the time to first acute exacerbation (reported by the attending physician) between the nintedanib group and placebo group was observed (HR: 1.15; 95%CI: 0.54–2.42; *p* = 0.67), but in the INPULSIS-2 trial, the nintedanib group showed significantly prolonged survival compared with the placebo group (HR: 0.38; 95%CI: 0.19–0.77; *p* = 0.005). In addition, combined analysis of the INPULSIS-1 and INPULSIS-2 trials showed no significant difference in the time to the first acute exacerbation (reported by the attending physician) (HR: 0.64, 95%CI: 0.39–1.05, *p* = 0.08). However, sensitivity analysis of acute exacerbation/suspicion of acute exacerbation by an independent committee using integrated data showed that the time until first acute exacerbation (HR: 0.32, 95%CI: 0.16–0.65, *p* = 0.001) and frequency of acute exacerbation were significantly different between the groups. In the INPULSIS-1 trial, the most common adverse event in the nintedanib group was diarrhea, and the incidence of diarrhea was 61.5% in the nintedanib group and 18.6% in the placebo group. In the INPULSIS-2 trial, the incidence of diarrhea in the nintedanib and placebo groups was 63.2% and 18.3%, respectively, but the severity was low.

Recently, a phase III international joint trial (SENSCIS trial), announced in 2019, targeting patients with interstitial lung disease associated with systemic scleroderma (SSc-ILD) showed nintedanib to be effective and safe for SSc-ILD patients [61]. This trial, the largest international, placebo-controlled, randomized, double-blind study on nintedanib was conducted in 576 patients with SSc-ILD in more than 32 countries, including the US, Canada, China, Japan, Germany, France, and the United Kingdom. The 52-week adjusted rate of annual reduction (mL/year) in FVC (mL) (the main endpoint) was −52.4 mL/year in the nintedanib group and −93.3 mL/year in the placebo group. The difference between groups was 41.0 mL/year (95%CI: 2.9–79.0; *p* = 0.04), which was similar to the results of the INPULSIS trials. The most common adverse event was diarrhea with an incidence of 75.7% in the nintedanib group and 31.6% in the placebo group; 49.5% of the incidences of diarrhea in the nintedanib group were mild, while 45% were moderate. As a result, nintedanib was approved by the FDA in the US in September 2019 and in Japan, in December 2019.

In an in vitro study, nintedanib inhibited angiogenesis and suppressed tumor cell proliferation; and as nintedanib is expected to be an antitumor drug, it is still under development [55]. A phase III trial (LUME-Lung 1 trial) to evaluate the efficacy and safety of the combination of docetaxel and nintedanib [62] showed that the median PFS (the primary endpoint) was significantly longer in the nintedanib plus docetaxel group than in the placebo plus docetaxel group (3.4 months vs. 2.7 months, HR: 0.79; 95%CI: 0.68–0.92, *p* = 0.0019). An analysis of 658 already diagnosed adenocarcinomas showed that the median OS in the nintedanib group was longer than that in the placebo group (12.6 months vs. 10.3 months, HR: 0.83, 95%CI: 0.70–0.99, *p* = 0.0359). The proportion of cases with squamous cell carcinomas was high in this study (42% of the total). Adverse events included diarrhea, liver dysfunction, nausea, loss of appetite, and vomiting. Grade 3 or higher adverse events were slightly higher in the nintedanib group than in the placebo group (71.3% vs. 64.3%), and grade 5 adverse events were higher in the nintedanib group than in the placebo group (16.4% vs. 11.8%). In the LUME-Lung 2 trial, which evaluated the effects of nintedanib and pemetrexed in previously treated NSCLC patients, PFS in the nintedanib group was significantly longer than that in the placebo group (4.4 months vs. 3.6 months, HR: 0.83, 95%CI: 0.7–0.99, *p* = 0.04), and the disease control rate in the nintedanib group was significantly better than that in the placebo group (61% vs. 53%, odds ratio 1.37, *p* = 0.039). However, as a result of the interim analysis, the registration was discontinued based on futility analysis of PFS evaluated by researchers [63]. Table 3 summarizes the results of clinical trials of sorafenib and nintedanib (Table 3).

## 7. Novel Compounds

Various studies on multitarget drugs for respiratory diseases are now in progress. Anlotinib, a relatively novel multitarget TKI for tumor angiogenesis and tumor cell proliferation, is effective as a third-line or beyond treatment for advanced NSCLC [64]. Entrectinib, another multitarget TKI of *TRKA*/*B*/*C*, *ROS1*, and *ALK*, has been studied in patients with advanced or metastatic solid tumors harboring *NTRK1*/*2*/*3*, *ROS1*, or *ALK* gene fusions [65].

However, some attention should be paid to multitarget therapies. Multitarget TKIs are used for NSCLC harboring *RET* rearrangement. Notably, these patients suffered from high-grade toxicity mainly induced by anti-VEGFR kinase activity. Therefore, selective *RET* inhibitors such as BLU-667, LOXO-292, and RXDX-105 have been recently investigated in early phase clinical trials and showed promising efficacy with a manageable toxicity profile [66].

## 8. Conclusions

We have summarized the available data regarding multitarget drugs used against respiratory diseases including lung cancer and IPF. In addition, various studies on multitarget drugs for respiratory diseases have just begun. Further advances in multitarget drugs will bring additional benefits to patients.

## Figures and Tables

**Figure 1 molecules-25-03987-f001:**
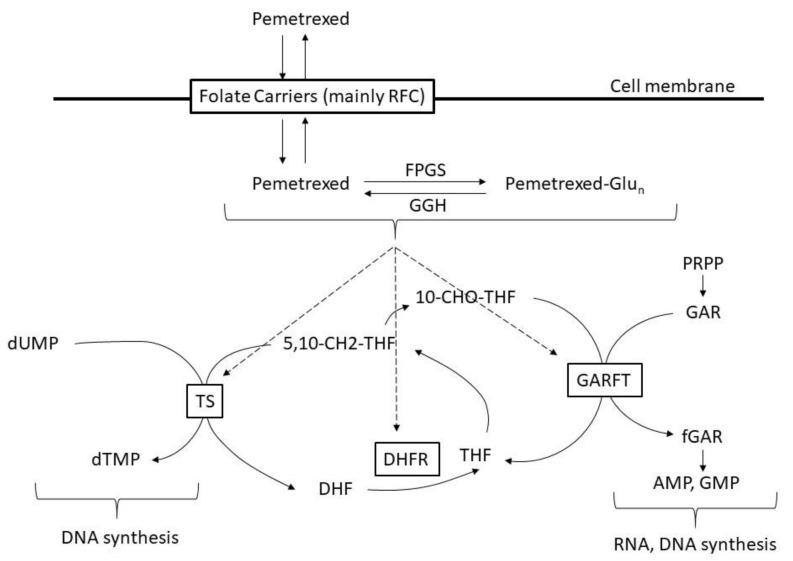
Mechanism of pemetrexed. RFC: reduced folate carrier, FPGS: folypoly-gamma-glutamate synthetase, GGH: gamma-glutamyl hydrolase, Glu_n_: glutamate, dUMP: deoxyyuridine monophosphate, dTMP: deoxythymidine monophosphate, 5,10-CH2-THF: 5,10-methenyl-tetrahydrofolate, DHF: dihydrofolate, THF: tetrahydrofolate, 10-CHO-THF: 10-formyl tetrahydrofolate, PRPP: phosphoribosyl pyrophosphate, GAR: glycinamide ribonucleotide, fGAR: formylglycinamide ribonucleotide, AMP: adenosine monophosphate, GMP: guanosine monophosphate.

**Figure 2 molecules-25-03987-f002:**
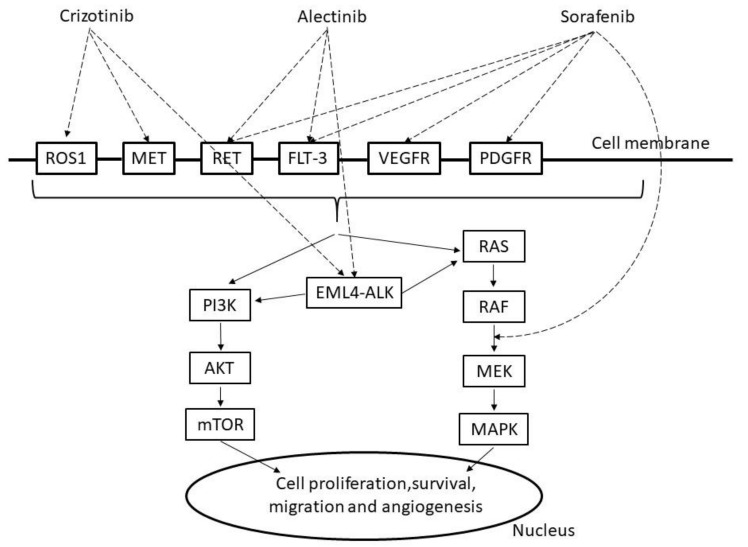
Mechanism of crizotinib, alectinib and sorafenib. ROS1: c-ROS oncogene1, MET: mesenchymal epithelial transition factor, RET: proto-oncogene ret, FLT-3: Fms-like tyrosine protein kinase, VEGFR: vascular endothelial growth factor receptor, PDGFR: platelet-derived growth factor receptors, PI3K: phosphoinositide 3 kinase, AKT: v-act murine thymoma viral oncogene homolog, mTOR: mammalian target of rapamycin, EML4-ALK: echinoderm microtubule associated protein like 4, RAS: rat sarcoma protein, RAF: v-raf murine viral oncogene homolog, MAPK: mitogen activated protein kinase, MEK:,MAPK/extracellular signal regulated kinase(ERK) kinase.

**Figure 3 molecules-25-03987-f003:**
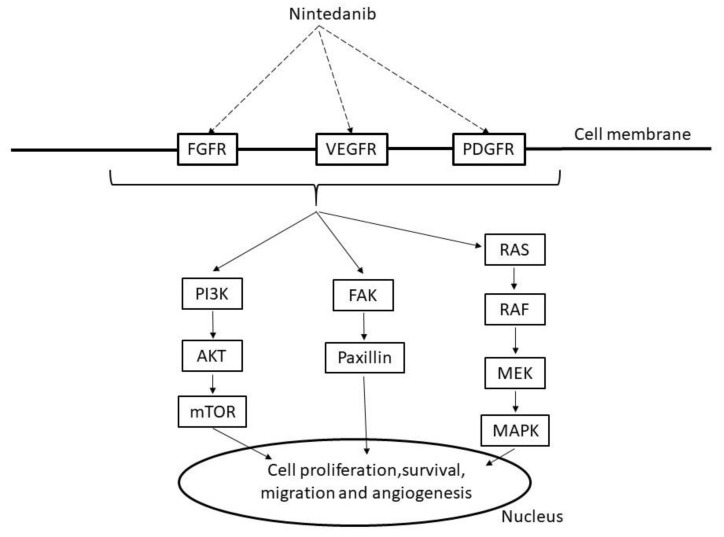
Mechanism of nintedanib. FGFR: fibroblast growth factor receptors, VEGFR: vascular endothelial growth factor receptor, PDGFR: platelet-derived growth factor receptors, PI3K: phosphoinositide 3 kinase, AKT: v-act murine thymoma viral oncogene homolog, mTOR: mammalian target of rapamycin, FAK: focal adhesion kinase, RAS: rat sarcoma protein, RAF: v-raf murine viral oncogene homolog, MAPK: mitogen activated protein kinase, MEK:,MAPK/extracellular signal regulated kinase(ERK) kinase.

**Table 1 molecules-25-03987-t001:** Results of clinical trials of pemetrexed.

Trial	Phase	Line	Type	n	RR(CR/PR)	DCR(CR/PR/SD)	Time to Progression	PFS	MST or OS
Scagliotti G.V. 2003 PEM alone	II	1st line	MPM	64	14.1%	65.6%	4.7 m		10.7 m
Vogelzang N.J. 2003 PEM/CDDP vs. CDDP	III	1st line	MPM	456	41.3% vs. 16.7%		5.7 m vs. 3.9 m		12.1 m vs. 9.3 m
Hughes A. 2002 PEM/CBDCA	I	1st line	MPM	27	32%	88%		305 d	451 d
Ceresoli G.L. 2006 PEM/CBDCA	II	1st line	MPM	102	18.6%	65.7%	6.5 m		12.7 m
Castagneto B. 2008 PEM/CBDCA	II	1st line	MPM	76	25%	63%	8 m		14.1 m
Hanna N. 2004 PEM vs. DTX	III	2nd line	NSCLC	571	9.1% vs. 8.8%		3.4 m vs. 3.5 m	2.9 m vs. 2.9 m	8.3 m vs. 7.9 m
Scagliotti G.V. 2008 CDDP/PEM vs. CDDP/GEM (JMDB trial)	III	1st line	NSCLC	1725	30.6% vs. 28.2%			4.8 m vs. 5.1 m	10.3 m vs. 10.3 m
			Non Sq	1000				5.3 m vs. 4.7 m	11.8 m vs. 10.4 m
			Sq	473				4.4 m vs. 5.5 m	9.4 m vs. 10.8 m
Paz-Ares L.G. 2013 PEM maintenance vs. placebo (PARAMOUNT trial)	III	Induction	NSCLC	939				4.4 m vs. 2.8 m	13.9 m vs. 11.0 m
Gandhi L. 2018 Platinum/PEM/ Pembrolizumab vs. Platinum/PEM/ placebo	III	1st line	NSCLC	616	47.6% vs. 18.9%	84.6% vs. 70.4%		8.8 m vs. 4.9 m	Not reachedvs 11.3 m

MPM: malignant pleural mesothelioma, NSCLC: non-small cell lung cancer, NonSq: nonsquamous cell carcinoma, RR: response rate, CR: complete response, PR: partial response, DCR: disease control rate, SD: stable disease, PFS: progression free survival, MST: medial survival time, OS: overall survival, PEM: pemetrexed, CDDP: cisplatin, CBDCA: carboplatin, DTX: docetaxel, GEM: gemcitabine.

**Table 2 molecules-25-03987-t002:** Results of clinical trials of crizotinib and alectinib.

	Trial	Phase	Line	Type	n	RR(CR/PR)	DCR(CR/PR/SD)	PFS	MST or OS
Crizotinib	Kwak E.L. 2010 (PROFILE 1001)	I	1st line	*ALK*-positive NSCLC	82	57%	90%	6.4 m	
	Shaw A.T. 2013 Crizotinib vs. PEM or DTX (PROFILE 1007)	III	2nd line	*ALK*-positive LC	347	65% vs. 20%	84% vs. 56%	7.7 m vs. 3.0 m	20.3 m vs. 22.8 m
	Solomon B.J. 2014 Crizotinib vs. PEM/platinum (PROFILE 1014)	III	1st line	*ALK*-positive NSCLC(NonSq)	343	74% vs. 45%	91% vs. 82%	10.9 m vs. 7.0 m	Not reached
	Shaw A.T. 2014 Crizotinib alone	I	1st line	*ROS-1*-positive NSCLC	50	72%	90%	19.2 m	Not reached
	Wu Y-L. 2018 Crizotinib alone	II	4th line or later	*ROS-1*-positive NSCLC	127	71.7%	80.3%(16 w)	15.9 m	32.5 m
Alectinib	Seto T. 2013 Alectinib alone (AF-001JP study)	I	3rd line or later	*ALK*-positive NSCLC(ALK inhibitor-naïve)	24				
		II	2nd line or later		46	93.5%	95.7%		
	Gadgeel S.M. 2014 (AF-002JG trial) Alectinib alone	I/II	any	ALK-positive NSCLC(crizotinib-treated)	47	55%	91%		
				With CNS metastases	21	52%	90%		
	Hida T. 2017 Alectinib vs. crizotinib (J-ALEX trial)	III	1–2nd line	ALK-positive NSCLC(ALK inhibitor-naïve Japanese)	207	92% vs. 79%	96% vs. 92%	Not reached vs. 10.2 m	Not reached
	Nakagawa K. 2020 Alectinib vs. crizotinib (J-ALEX trial)	III	1–2nd line	ALK-positive NSCLC(ALK inhibitor-naïve Japanese)	207			34.1 m vs. 10.2 m	Not reached vs. 43.7 m
	Peters S. 2017 Alectinib vs. crizotinib (ALEX trial)	III	1st line	ALK-positive NSCLC	303	82.9% vs. 75.5%	89% vs. 91%	Not reached vs. 11.1 m	Not reached

NSCLC: non-small cell lung cancer, NonSq: nonsquamous cell carcinoma, CNS: central nervous system, RR: response rate, CR: complete response, PR: partial response, DCR: disease control rate, SD: stable disease, PFS: progression free survival, MST: medial survival time, OS: overall survival, PEM: pemetrexed, DTX: docetaxel.

**Table 3 molecules-25-03987-t003:** Results of clinical trials of sorafenib and nintedanib.

	Trial	Phase	Line	Type	n	RR(CR/PR)	DCR(CR/PR/SD)	Time to progression	PFS	MST or OS
Sorafenib	Blumenschein G.R. 2009 Sorafenib alone	II	2nd–3rd line	NSCLC	54	0%	59%		2.7 m	6.7 m
	Paz-Ares L. 2015 Sorafenib alone vs. placebo (MISSION trial)	III	3rd–4th line	NSCLC	703	4.9% vs. 0.9%	47.1% vs. 24.7%	2.9 m vs. 1.4 m	2.8 m vs. 1.4 m	8.2 m vs. 8.3 m
				EGFR mutation+	89	6.8% vs. 0%	40.9% vs. 2.2%		2.7 m vs. 1.4 m	13.9 m vs. 6.5 m
				wild-type EGFR	258	7.4% vs. 1.5%	46.7% vs. 25.8%		2.7 m vs. 1.5 m	8.3 m vs. 8.4 m
				KRAS mutation+	68	2.9% vs. 0%	44.1% vs. 7.6%		2.6 m vs. 1.7 m	6.4 m vs. 5.1 m
				wild-type KRAS	279	8.3% vs. 1.4%	45.4% vs. 20.4%		2.7 m vs. 1.4 m	11.0 m vs. 9.1 m
	Scagliotti G. 2010 CBDCA/PTX/sorafenib vs. CBDCA/PTX/placebo (ESCAPE trial)	III	1st line	NSCLC	926	27% vs. 24%	50% vs. 56%		4.6 m vs. 5.4 m	10.7 m vs. 10.6 m
				Sq	223	25% vs. 35%	42% vs. 60%		4.3 m vs. 5.8 m	8.9 m vs. 13.6 m
				Other	703	28% vs. 20%	52% vs. 55%		4.8 m vs. 5.3 m	11.5 m vs. 10.2 m
Nintedanib	Reck M. 2014 DTX/nintedanib vs. DTX/placebo (LUME-Lung 1 trial)	III	2nd line	NSCLC	1314	4.4% vs. 3.3%	54% vs. 41.3%		3.4 m vs. 2.7 m	10.1 m vs. 9.1 m
				Adeno	658	4.7% 3.6%	60.2% vs. 44%			12.6 m vs. 10.3 m
	Hanna N.H. 2016 Nintedanib/PEM vs. PEM (LUME-Lung 2 trial)	III	2nd line	NonSq NSCLC	713	9.1% vs. 8.3%	60.9% vs. 53.3%		4.4 m vs. 3.6 m	12.0 m vs. 12.7 m
				Adeno	670	9.6% vs. 9.0%	61.8% vs. 54.6%		4.5 m vs. 3.9 m	12.3 m vs. 13.1 m

NSCLC: non-small cell lung cancer, NonSq: nonsquamous cell carcinoma, RR: response rate, CR: complete response, PR: partial response, DCR: disease control rate, SD: stable disease, PFS: progression free survival, MST: medial survival time, OS: overall survival, EGFR: epidermal growth factor receptor, KRAS: kirsten rat sarcoma viral oncogene homolog, CBDCA: carboplatin, PTX: paclitaxel, DTX: docetaxel, PEM: pemetrexed.

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
