# Peer review of "Novel Multitarget Therapies for Lung Cancer and Respiratory Disease"

_molecules, 2020, doi:10.3390/molecules25173987_

Round 1

Reviewer 1 Report

Authors have nicely discussed the use of multitarget drugs against respiratory diseases pointing out the mechanisms, opposing the indication of single-target drug use, based on its ineffectiveness, or the prescription of the combination of drugs because of the increased possibility of causing side effects. The tables are very informative. They made a very good point claiming further advances in multitarget drugs will be beneficial. Just one matter needs attention to clarify the physiopathology and involved mechanisms. Figure 1 is good. Make figures for each mentioned drug and their mechanisms of action or one figure/table containing all the mentioned drugs and their targets and the biological effect of each target activation. More detailed information about the targets is missing. The meaning of some abbreviation is also missing.

Author Response

At first, we really thank you for constructive comments that help us to improve the quality of our presentation. We hope that the changes we made are enough to make this manuscript acceptable for publication in molecules. We agreed your comments and added figures and the descriptions about targets and abbreviation. The revised text was highlighted in red.

Reviewer 2 Report

In this manuscript, the authors provide a short summary on the mechanisms of action of multi-target therapies and report the results of the latest clinical trials. The manuscript is well written and the illustration is presented in a good quality. This will provide interesting information for the reader of the journal. However, there are still many grammatical and syntax errors in the article. So I think the manuscript can be published after grammar and language check.

Author Response

We really thank for your time despite your busyness. We thank this comment and corrected our text by English proofread service. The revised version was created using the track change mode in MS-Word to show the changes made.

Round 2

Reviewer 1 Report

The authors answered all matters. The newly added figures are excellent and made the review clearer.